# Preoperative Chronic Inflammation Is a Risk Factor for Postoperative Complications Independent of Body Composition in Gastric Cancer Patients Undergoing Radical Gastrectomy

**DOI:** 10.3390/cancers16040833

**Published:** 2024-02-19

**Authors:** Ryota Matsui, Noriyuki Inaki, Toshikatsu Tsuji, Tetsu Fukunaga

**Affiliations:** 1Department of Gastroenterological Surgery, Ishikawa Prefectural Central Hospital, Kanazawa 920-8201, Japan; toshi_toshi25@yahoo.co.jp; 2Department of Upper Gastrointestinal Surgery, Juntendo University Hospital, Tokyo 113-8431, Japan; t2fukunaga@juntendo.ac.jp; 3Department of Gastroenterological Surgery, The Cancer Institute Hospital of Japanese Foundation for Cancer Research, Tokyo 135-8550, Japan; 4Department of Gastrointestinal Surgery/Breast Surgery, Graduate School of Medical Science, Kanazawa University, Kanazawa 920-8530, Japan

**Keywords:** body composition, gastrectomy, gastric cancer, inflammation, postoperative complication

## Abstract

**Simple Summary:**

Patients with gastric cancer who underwent gastrectomy after surgery had a poor long-term prognosis when postoperative complications occurred. However, the relationship between preoperative inflammation and postoperative complications in patients with gastric cancer who underwent gastrectomy remains unclear. For example, it has been suggested that intervention methods may differ depending on whether chronic inflammation is present in patients with malnutrition. This study aimed to determine the relationship between preoperative mild inflammation and postoperative complications in patients with gastric cancer after gastrectomy. As a result, management to ameliorate inflammation is necessary if preoperative chronic inflammation is present.

**Abstract:**

The purpose of this study was to investigate the association between preoperative inflammation and postoperative complications in gastric cancer patients having elective gastrectomy. Participants in this study were those who underwent radical gastrectomy between April 2008 and June 2018 and were diagnosed with stage I–III primary gastric cancer. Preoperative CRP values were used to divide the patients into two groups: the inflammation group comprised individuals having a CRP level of ≥0.5 mg/dL; the other was the non-inflammation group. The primary outcome was overall complications of Clavien–Dindo grade II or higher after surgery. Using propensity score matching to adjust for background, we compared the postoperative outcomes of the groups and conducted a multivariate analysis to identify risk variables for complications. Of 951 patients, 852 (89.6%) were in the non-inflammation group and 99 (10.4%) were in the inflammation group. After matching, both groups included 99 patients, and no significant differences in patient characteristics were observed between both groups. The inflammation group had a significantly greater total number of postoperative complications (*p* = 0.019). The multivariate analysis revealed that a preoperative CRP level of ≥0.5 mg/dL was an independent risk factor for total postoperative complications in all patients (odds ratio: 2.310, 95% confidence interval: 1.430–3.730, *p* < 0.001). In conclusion, in patients undergoing curative resection for gastric cancer, preoperative inflammation has been found to be an independent risk factor for overall complications after surgery. Patients with chronic inflammation require preoperative treatment to reduce inflammation because chronic inflammation is the greatest risk factor for postoperative complications.

## 1. Introduction

The occurrence of postoperative complications is associated with a poor long-term prognosis in patients with gastric cancer who have undergone gastrectomy [1,2]. High visceral fat mass and low muscle mass have been demonstrated to increase postoperative complications owing to a high postoperative inflammatory response [3,4]. High levels of inflammatory cytokines, typically interleukin-6, on the first postoperative day have been shown to increase postoperative complications [5]. By contrast, immunonutrition administered preoperatively to reduce excessive postoperative inflammatory reactions has been reported to decrease postoperative complications [4]. This is one piece of evidence that reducing excessive postoperative inflammation decreases postoperative complications. Therefore, treatment to reduce inflammation during the preoperative period is necessary.

It is yet unknown how preoperative inflammation and postoperative complications relate to patients with gastric cancer undergoing gastrectomy. In elective gastrectomy, postoperative inflammation reflects acute inflammation caused by surgical invasion, whereas preoperative inflammation is expected to reflect chronic inflammation such as cachexia, except in the setting of acute inflammation. In patients with acute inflammation, elective surgery is usually canceled. However, in patients with chronic inflammation, surgery is often performed as scheduled. This reflects the fact that mild inflammation is considered to have no impact on surgery or postoperative course. If chronic inflammation adversely affects the postoperative course, surgical cancellation or action to reduce inflammation may be necessary. Preoperative immunomodulatory nutrition is one option to reduce preoperative inflammation. Therefore, it is necessary to clarify the impact of chronic inflammation on the postoperative course in order to consider these indications.

This study aimed to determine the relationship between preoperative inflammation and postoperative complications in patients with gastric cancer undergoing elective gastrectomy. We hypothesized that preoperative chronic inflammation increases the incidence of postoperative complications after gastrectomy.

## 2. Materials and Methods

### 2.1. Eligibility Criteria

We conducted this retrospective case-control study at Ishikawa Prefectural Central Hospital. Participants in the study were those who, between April 2008 and June 2018, underwent a radical gastrectomy and were diagnosed with stage I–III primary gastric cancer according to the 15th edition of the Japanese Classification of Gastric Carcinoma [6]. As exclusion criteria, we excluded patients with residual gastric cancer, cancer in other organs, distant metastases, and acute inflammation as a comorbidity. By using the hospital’s electronic patient record system, we collected clinical and laboratory data as well as medical records and images.

In accordance with the Ishikawa Prefectural Central Hospital Institutional Ethical Review Committee’s approval of all experimental protocols (authorization number: 1895), the study followed the ethical guidelines outlined by the Ministry of Health, Labour, and Welfare for Medical and Health Research Involving Human Subjects, along with the Declaration of Helsinki. All patients were provided the opportunity to decline participation via an opt-out recruitment method.

### 2.2. Data Collection and Definition

Patients were categorized into two groups based on their preoperative C-reactive protein (CRP) levels: those with CRP levels ≥ 0.5 mg/dL were considered the inflammation group, and those without inflammation were considered the non-inflammation group.

Within 1 month postoperatively, computed tomography images were used along with the Ziostation software (ZIOSOFT, Tokyo, Japan) to measure body composition. The skeletal muscle mass index (SMI) was calculated by dividing the L3 cross-sectional area by the height squared to determine muscle mass. Meanwhile, visceral fat mass was measured based on the cross-sectional area of a single slice at the umbilical level. We used the previous Asian cut-off values [7] to set the cut-off values for men and women separately.

In this research, malnutrition and its severity were determined according to the criteria set by the Global Leadership Initiative on Malnutrition (GLIM) [8]. The diagnosis utilized the body mass index (BMI) and weight loss rate in individuals identified as at risk through nutritional screening using the Subjective Global Assessment (SGA). Specifically, moderate malnutrition was defined as a weight loss rate of 5–10% within the past 6 months or 10–20% beyond 6 months, or a BMI of <20.0 kg/m^2^ if <70 years old or <22.0 kg/m^2^ if ≥70 years old. Severe malnutrition was identified as a weight loss rate of >10% within the past 6 months or >20% beyond 6 months, or a BMI of <18.5 kg/m^2^ if <70 years old or <20.0 kg/m^2^ if >70 years old. Eligible patients were categorized into three groups: normal nutrition group, moderate malnutrition group, and severe malnutrition group. Those without malnutrition were classified as normal. Notably, we did not include SMI in the GLIM criteria, and its impact on postoperative complications was separately investigated.

We characterized chronic kidney disease (CKD) as an estimated glomerular filtration rate (eGFR) < 60 mL/min/1.73 m^2^, diabetes as a history of treatment or preoperative HbA1c ≥6.5%, chronic obstructive pulmonary disease (COPD) as FEV1.0% < 70% on spirometry, and congestive heart failure (CHF) as either a history of treatment or an ejection fraction of less than 50% on echocardiogram.

### 2.3. Outcomes

The study’s main focus was on overall postoperative complications. Secondary endpoints encompassed severe complications, infectious complications, operation time, intraoperative blood loss, and the duration of postoperative hospitalization. Postoperative complications were stratified according to Clavien–Dindo (CD) grade II or higher within the initial 30 days following surgery [9]. Severe complications were specifically defined as CD classification grade IIIa or higher. The outcomes of the two groups were subjected to comparison, with adjustments made for variations in background characteristics. The surgical site infections (SSIs) included incisional SSI and intra-abdominal infection.

### 2.4. Statistical Analyses

We categorized patients into inflammation and non-inflammation groups and adopted propensity score matching (PSM) with the aim of eliminating confounding factors in the comparison of various postoperative outcomes. A logistic regression model was utilized to determine the propensity score, incorporating covariates such as age, sex, surgical approach, surgical procedure, clinical stage, and comorbidities. The nearest neighbor matching method was applied, achieving one-to-one matching with a caliper size of 0.20. Following matching, comparisons of patient characteristics and postoperative outcomes were conducted using the Mann–Whitney U-test for continuous variables and the chi-squared or Fisher’s exact test for categorical variables. Multivariate analyses, employing logistic regression, were conducted to identify risk factors for postoperative complications and calculate odds ratios (ORs). All statistical analyses were performed using EZR software Ver.1.64 (Saitama Medical Center, Jichi Medical University, Saitama, Japan), with the significance level set at *p* < 0.05.

## 3. Results

### 3.1. Baseline Characteristics

Figure 1 presents the study flowchart, and Table 1 displays a comparison of patient characteristics. Of the 951 patients, 852 (89.6%) were in the non-inflammation group, and 99 (10.4%) were in the inflammation group. The inflammation group had a higher age (*p* < 0.001), more advanced clinical stage (*p* = 0.002), higher open surgery rate (*p* < 0.001), and higher total gastrectomy rate (*p* = 0.019) than the non-inflammation group. However, after matching, both groups included 99 patients, without significant differences in patient characteristics.

### 3.2. Postoperative Results Compared before and after Matching

Table 2 displays the results of the comparative analysis of postoperative outcomes between the inflammation and non-inflammation groups. Before matching, the inflammation group had significantly higher intraoperative blood loss (*p* = 0.001) and longer postoperative hospital stay (*p* = 0.003) compared with the non-inflammation group. The overall incidence of postoperative complications was significantly higher in the inflammation group (*p* = 0.001), including a significantly greater occurrence of infectious complications (*p* = 0.007), postoperative pneumonia (*p* = 0.011), and anastomotic leakage (*p* = 0.020), in comparison with the non-inflammation group.

After matching, there were no statistically significant differences in the operation time and intraoperative blood loss between the two groups. However, the inflammation group had a significantly longer postoperative hospital stay (*p* = 0.016) and significantly greater total number of postoperative complications (*p* = 0.019), which was the primary outcome of the study, than the non-inflammation group. Although infectious complications were more common in the inflammation group than in the non-inflammation group, the difference between the two groups was not significant (*p* = 0.067). Additionally, no significant differences in severe complications were observed between the two groups (*p* = 0.459).

### 3.3. Multivariate Analysis Related to Postoperative Complications

Table 3 presents the results of the multivariate analysis using the forward stepwise procedure to identify the risk factors for total postoperative complications. The analysis revealed that being male (OR: 1.820, 95% confidence interval [CI]: 1.190–2.770, *p* = 0.006), having a preoperative CRP level of ≥0.5 mg/dL (OR: 2.310, 95% CI: 1.430–3.730, *p* < 0.001), and having a visceral fat area ≥ 100 cm^2^ (OR: 1.680, 95% CI: 1.180–2.380, *p* = 0.004) were significant independent risk factors for total postoperative complications.

### 3.4. Multivariate Analysis Related to Postoperative Complications in Patients without Chronic Inflammation

Table 4 presents the results of the multivariate analysis to identify the risk factors for total postoperative complications in patients without chronic inflammation. The analysis revealed that being male (OR: 2.120, 95% CI: 1.330–3.370, *p* = 0.001) and having a visceral fat area ≥ 100 cm^2^ (OR: 1.690, 95% CI: 1.140–2.510, *p* = 0.009) were significant independent risk factors for total postoperative complications.

### 3.5. Multivariate Analysis Related to Postoperative Complications in Patients with Chronic Inflammation

Table 5 displays the outcomes of the multivariate analysis aimed at identifying risk factors for overall postoperative complications in patients with chronic inflammation. Multivariate analysis did not detect any significant risk factors in patients with chronic inflammation.

## 4. Discussion

To our knowledge, this study represents the first evidence indicating that preoperative mild inflammation serves as a risk factor for postoperative complications following radical gastrectomy in patients with gastric cancer. We compared the postoperative outcomes in the two groups using PSM with background adjustment and observed an increase in the total number of postoperative complications in the inflammation group. Furthermore, through a multivariate analysis that included all the patients, preoperative mild inflammation with CRP levels ≥ 0.5 mg/dL was identified as an independent risk factor for postoperative complications. This study highlights the need for intervention to reduce preoperative mild inflammation.

In previous reports, excessive postoperative inflammation has been documented to increase the incidence of postoperative complications [5,10,11,12,13]. Increased inflammation immediately postoperatively has been reported to be associated with a higher incidence of postoperative complications, and the CRP level has been identified as a factor involved in systemic immune-inflammation postoperatively [13]. In addition to CRP as an inflammatory marker, other markers such as IL-6 and presepsin have been shown to correlate with postoperative complications [5,10,11,12,13]. Regarding the breakdown of complications, it has been reported that high postoperative CRP in gastric cancer, esophageal cancer, and colorectal cancer is associated with an increased risk of anastomotic leakage [14,15]. Therefore, the suppression of inflammatory responses has been highlighted as one of the strategies for preventing postoperative complications.

This study showed that preoperative mild inflammation was an independent risk factor for the total number of postoperative complications. In this study, we set the cut-off value for CRP levels at 0.5 mg/dL; this value is widely used as an indicator of inflammation in many facilities in Japan. Many facilities in Japan consider more than 0.5 mg/dL to be an abnormal laboratory value. Therefore, using this cut-off value to determine preoperative inflammation may reflect the clinical results of many facilities in Japan. Furthermore, the Glasgow prognostic score, which has been used as a prognostic indicator, also adopts a cut-off value of 0.5 mg/dL [16,17,18]. The potential to predict not only the prognosis but also postoperative complications was demonstrated using this value preoperatively. In this study, 10% of the patients were assigned to the inflammation group using this cut-off value. As this study excluded patients with preoperative acute inflammation, the inflammation in this study may reflect the inflammation caused by cancer-induced cachexia [19,20]. Hence, the inflammation group had a greater proportion of patients with advanced clinical stage disease, which required adjustment by PSM, than the non-inflammation group. For advanced cancer patients with inflammation, anti-inflammatory agents such as non-steroidal anti-inflammatory drugs (NSAIDs) or aspirin, as well as immunomodulatory nutritional supplements like n-3 fatty acids, may be necessary for inflammation control [4].

The mechanism by which preoperative inflammation increased postoperative complications was unclear in this study. Acute inflammation simultaneously produces an immunosuppressive response, and the persistence of inflammation leads to compensatory anti-inflammatory response syndrome, which triggers infectious states such as sepsis [21]. This is explained by the occurrence of anti-inflammatory cytokines in response to the production and mobilization of proinflammatory cytokines, and the balance between the two has been reported to be associated with infectious complications [22,23]. Surgical procedures can potentially cause endothelial dysfunction, neutrophil activation, and systemic inflammation through the secretion of proinflammatory cytokines, and the risk of organ dysfunction increases if inflammation persists [22]. This could also be explained as a potential factor that increases postoperative complications due to chronic inflammation. Even though it is unclear as to whether the high postoperative inflammatory response is the cause or the effect of complications, identifying preoperative inflammation is still advantageous as an intervention target because it is recognizable.

Body composition analysis has attracted attention as one of the indicators for predicting postoperative inflammatory response. As targets prone to high levels of postoperative inflammatory mediators, sarcopenia with low muscle mass and excess visceral fat volume has been identified [3,4]. Inflammation in adipose tissue is reported to occur when dysfunctional adipocytes secrete inflammatory cytokines, which can negatively impact distant organ function [24]. Furthermore, sarcopenia and increased visceral fat mass have been reported to lead to chronic inflammation [25,26]. Therefore, we investigated these body compositions. In addition to preoperative inflammation, high visceral fat mass was identified as an independent risk factor for postoperative complications. Systematic review shows that high visceral fat mass increases postoperative complications in patients with upper gastrointestinal cancer [27]. Therefore, body composition serves as a useful indicator for predicting postoperative inflammation, and interventions to suppress inflammation may be necessary for patients with a greater amount of visceral fat.

The breakdown of complications demonstrated a tendency toward infectious complications and anastomotic leakage. The former may result from immunosuppression due to persistent chronic inflammation, and the latter may be due to inflammation delaying wound healing. High postoperative inflammation has been reported as a risk factor for anastomotic leakage after gastric, esophageal, and colorectal resection [14,15,28,29]. The relationship between inflammation and delayed wound healing is unclear; however, prolonged preoperative inflammation may also be a risk factor for anastomotic leakage. Further studies are needed to clarify the mechanism by which inflammation increases anastomotic leakage. Given the elevated rate of postoperative complications, the inflammation group experienced a significantly extended duration of hospital stay compared with the non-inflammation group.

One of the strategies to suppress inflammation is the use of immunomodulatory nutritional supplements. Immunonutrition has the capability to quell excessive inflammation [30]. A study demonstrated that preoperative immunonutrition effectively subdued inflammatory cytokines following pancreaticoduodenectomy, potentially contributing to a reduction in postoperative complications [4]. Another study indicated that administering immunonutrition postoperatively led to a decrease in postoperative CRP levels [31]. These findings suggest that both pre- and postoperative administrations of immunonutrition may mitigate inflammation. Additionally, it serves as immunostimulation against inflammation following surgical invasion or trauma, which induces immunosuppression and diminishes resistance to infection [32]. The former is recommended for preoperative use as a modulator, while the latter is recommended for postoperative use as a stimulator [32]. Further studies are needed to determine whether immunonutrition reduces postoperative complications in gastric cancer patients with preoperative mild inflammation.

The study has several limitations. Firstly, it was a single-center retrospective study. To validate the generalizability of the results, a prospective multicenter study utilizing the same cut-off values should be undertaken. Secondly, the mechanism linking preoperative inflammation to postoperative complications remains unclear. Further investigations are necessary to delve into the underlying causes of this phenomenon. Thirdly, we cannot identify the cause of chronic inflammation. We cannot rule out other effects of chronic disease besides those arising from the cancer itself. Fourthly, the preoperative inflammatory status is assessed at a single time point, and there is no mention of the duration of sustained inflammation. Since exercise and immunonutrition have demonstrated efficacy in reducing inflammation [4,29], a combination of these approaches may prove effective in preventing postoperative complications in individuals with preoperative inflammation. Our goal was to assess the efficacy of nutritional interventions, such as immunonutrition, in improving postoperative outcomes for patients with preoperative mild inflammation.

## 5. Conclusions

Preoperative inflammation has been identified as a risk factor for postoperative complications in patients undergoing curative resection for gastric cancer. The multivariate analysis indicated preoperative inflammation as an independent risk factor for postoperative complications. Patients with chronic inflammation require preoperative treatment to reduce inflammation because chronic inflammation is the greatest risk factor for postoperative complications.

## Figures and Tables

**Figure 1 cancers-16-00833-f001:**
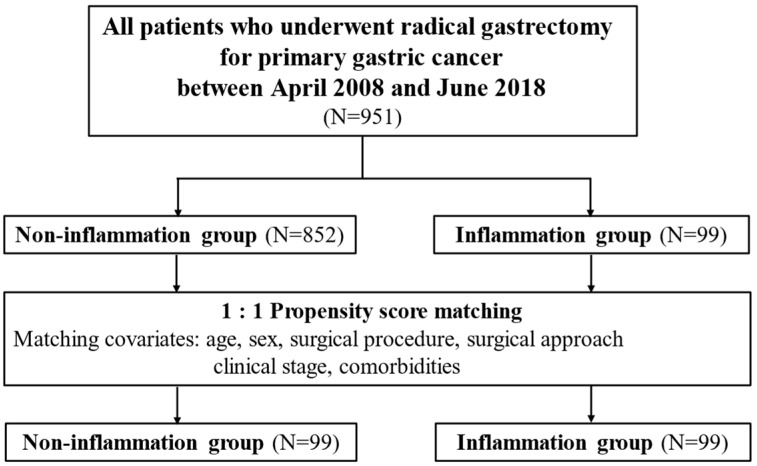
Study design.

**Table 1 cancers-16-00833-t001:** Patient characteristics before and after propensity score matching.

	All Patients			After Matching		
	Non-Inflammation Group (N = 852)	Inflammation Group (N = 99)	*p* Value	Non-Inflammation Group (N = 99)	Inflammation Group (N = 99)	*p* Value
Sex			0.307			1.000
Male	574 (67.4%)	72 (72.7%)		72 (72.7%)	72 (72.7%)	
Female	278 (32.6%)	27 (27.3%)		27 (27.3%)	27 (27.3%)	
Age, mean ± SD	65.81 ± 11.13	71.28 ± 9.66	<0.001	71.71 ± 8.84	71.28 ± 9.66	0.747
Body mass index, mean ± SD	22.95 ± 3.26	22.83 ± 3.72	0.743	23.21 ± 3.37	22.83 ± 3.72	0.458
Surgical approach			<0.001			0.773
Laparoscopic surgery	668 (78.4%)	57 (57.6%)		60 (60.6%)	57 (57.6%)	
Open surgery	184 (21.6%)	42 (42.4%)		39 (39.4%)	42 (42.4%)	
Surgical procedure			0.019			0.367
Distal gastrectomy	576 (67.6%)	57 (57.5%)		54 (54.5%)	57 (57.5%)	
Proximal gastrectomy	78 (9.2%)	6 (6.1%)		12 (12.1%)	6 (6.1%)	
Total gastrectomy	198 (23.2%)	36 (36.4%)		33 (33.3%)	36 (36.4%)	
Lymph node dissection			0.914			0.665
D1+	510 (59.8%)	60 (60.6%)		56 (56.6%)	60 (60.6%)	
D2	342 (40.2%)	39 (39.4%)		43 (43.4%)	39 (39.4%)	
Clinical stage			0.002			0.857
I	573 (67.3%)	51 (51.5%)		50 (50.5%)	51 (51.5%)	
II	135 (15.8%)	17 (17.2%)		15 (15.2%)	17 (17.2%)	
III	144 (16.9%)	31 (31.3%)		34 (34.3%)	31 (31.3%)	
Comorbidity						
CKD	137 (16.1%)	20 (20.2%)	0.316	24 (24.2%)	20 (20.2%)	0.608
COPD	174 (20.4%)	29 (29.3%)	0.051	30 (30.3%)	29 (29.3%)	1.000
Diabetes	129 (15.1%)	21 (21.2%)	0.144	13 (13.1%)	21 (21.2%)	0.187
SMI (cm^2^/m^2^)						
Median (IQR)	40.01 (34.50–46.82)	40.50 (34.31–44.80)	0.378	38.75 (33.96–44.51)	40.50 (34.31–44.80)	0.833
Low-SMI	337 (39.6%)	45 (45.5%)	0.503	42 (42.4%)	45 (45.5%)	0.900
VFA (cm^2^/m^2^)						
Median (IQR)	78.95 (41.28–128.9)	87.55 (49.30–125.0)	0.237	66.35 (34.55–133.4)	87.55 (49.30–125.0)	0.190
≥100 cm^2^/m^2^	298 (38.1%)	40 (43.0%)	0.369	31 (34.1%)	40 (43.0%)	0.229
GLIM-defined malnutrition	221 (25.9%)	37 (37.4%)	0.023	28 (28.3%)	37 (37.4%)	0.226
Moderate	124 (14.6%)	19 (19.2%)	0.234	14 (14.1%)	19 (19.2%)	0.446
Severe	97 (11.4%)	18 (18.2%)	0.071	14 (14.1%)	18 (18.2%)	0.563

**Table 2 cancers-16-00833-t002:** Comparison of postoperative outcomes.

	Before Matching			After Matching		
	Non-Inflammation Group (N = 852)	Inflammation Group (N = 99)	*p* Value	Non-Inflammation Group (N = 99)	Inflammation Group (N = 99)	*p* Value
Operation time (min), median (IQR)	250 (210, 310)	235 (198, 300)	0.123	240 (195, 310)	235 (198, 300)	0.738
Intraoperative blood loss (g), median (IQR)	20.0 (10.0, 50.0)	30.0 (10.0, 147.5)	0.001	30.0 (10.0, 150.0)	30.0 (10.0, 147.5)	0.450
Postoperative hospital stay (days), median (IQR)	14.0 (11.0, 18.0)	16.0 (12.0, 25.5)	0.003	15.0 (10.5–19.5)	16.0 (12.0–25.5)	0.016
**Postoperative complications**						
Infectious complications	115 (13.5%)	24 (24.2%)	0.007	13 (13.1%)	24 (24.2%)	0.067
Pneumonia	18 (2.1%)	7 (7.1%)	0.011	4 (4.0%)	7 (7.1%)	0.537
Incisional SSI	25 (2.9%)	3 (3.0%)	1.000	1 (1.0%)	3 (3.0%)	0.621
Intra-abdominal abscess	59 (6.9%)	10 (10.1%)	0.303	4 (4.0%)	10 (10.1%)	0.164
Pancreatic fistula	41 (4.8%)	2 (2.0%)	0.305	4 (4.0%)	2 (2.0%)	0.683
Anastomotic leakage	35 (4.1%)	10 (10.1%)	0.020	3 (3.0%)	10 (10.1%)	0.082
Severe complications	51 (6.0%)	11 (11.1%)	0.081	7 (7.1%)	11 (11.1%)	0.459
Total complications	145 (17.0%)	31 (31.3%)	0.001	16 (16.2%)	31 (31.3%)	0.019

**Table 3 cancers-16-00833-t003:** Results of multivariate analysis of risk factors for total postoperative complications in all patients.

Variables	Univariate Analysis			Multivariate Analysis		
	OR	95% CI	*p* Value	OR	95% CI	*p* Value
Sex						
Female	1			1		
Male	2.250	1.500–3.360	<0.001	1.840	1.210–2.800	0.004
Age (years)						
<70	1			1		
≥70	1.340	0.960–1.860	0.086	1.140	0.799–1.640	0.461
Surgical procedure						
Distal gastrectomy	1			1		
Total gastrectomy	1.460	1.020–2.100	0.039	1.380	0.949–2.000	0.093
Surgical approach						
Laparoscopic surgery	1					
Open surgery	1.350	0.935–1.950	0.110			
Lymph node dissection						
D1+	1					
D2	0.983	0.704–1.370	0.921			
Clinical stage						
I	1					
≥II	1.220	0.868–1.710	0.255			
CKD						
Absent	1			1		
Present	1.730	1.150–2.580	0.008	1.370	0.895–2.100	0.146
COPD						
Absent	1			1		
Present	1.610	1.110–2.340	0.012	1.380	0.930–2.040	0.111
Diabetes						
Absent	1			1		
Present	1.490	0.983–2.260	0.060	1.170	0.756–1.820	0.478
SMI						
High-SMI	1			1		
Low-SMI	0.737	0.521–1.040	0.086	0.869	0.599–1.260	0.457
Visceral fat area						
<100 cm^2^/m^2^	1			1		
≥100 cm^2^/m^2^	1.870	1.330–1.920	<0.001	1.560	1.080–2.250	0.017
GLIM-defined malnutrition						
Absent	1					
Present	0.906	0.624–1.320	0.606			
Preoperative CRP						
<0.5 mg/dL	1			1		
≥0.5 mg/dL	2.220	1.400–3.520	<0.001	1.950	1.210–3.160	0.006

**Table 4 cancers-16-00833-t004:** Results of multivariate analysis of risk factors for postoperative complications in patients without preoperative chronic inflammation.

Variables	Univariate Analysis			Multivariate Analysis		
	OR	95% CI	*p* Value	OR	95% CI	*p* Value
Sex						
Female	1			1		
Male	2.530	1.610–3.980	<0.001	2.120	1.330–3.370	0.001
Age (years)						
<70	1					
≥70	1.310	0.914–1.890	0.141			
Surgical procedure						
Distal gastrectomy	1					
Total gastrectomy	1.320	0.883–1.980	0.175			
Surgical approach						
Laparoscopic surgery	1					
Open surgery	1.310	0.862–1.980	0.209			
Lymph node dissection						
D1+	1					
D2	0.924	0.640–1.330	0.673			
Clinical stage						
I	1					
≥II	1.140	0.783–1.660	0.495			
CKD						
Absent	1			1		
Present	1.800	1.160–2.780	0.009	1.490	0.946–2.330	0.086
COPD						
Absent	1			1		
Present	1.490	0.985–2.260	0.059	1.330	0.867–2.030	0.193
Diabetes						
Absent	1			1		
Present	1.520	0.959–2.400	0.075	1.200	0.743–1.930	0.457
SMI						
High-SMI	1					
Low-SMI	0.725	0.493–1.060	0.100			
Visceral fat area						
<100 cm^2^/m^2^	1			1		
≥100 cm^2^/m^2^	2.100	1.440–3.050	<0.001	1.690	1.140–2.510	0.009
GLIM-defined malnutrition						
Absent	1					
Present	0.891	0.588–1.350	0.587			

**Table 5 cancers-16-00833-t005:** Results of multivariate analysis of risk factors for postoperative complications in patients with preoperative chronic inflammation.

Variables	Univariate Analysis			Multivariate Analysis		
	OR	95% CI	*p* Value	OR	95% CI	*p* Value
Sex						
Female	1					
Male	1.110	0.425–2.920	0.825			
Age (years)						
<70	1					
≥70	0.912	0.385–2.160	0.834			
Surgical procedure						
Distal gastrectomy	1					
Total gastrectomy	1.720	0.721–4.110	0.221			
Surgical approach						
Laparoscopic surgery	1					
Open surgery	0.971	0.411–2.300	0.947			
Lymph node dissection						
D1+	1					
D2	1.420	0.598–3.350	0.429			
Clinical stage						
I	1					
≥II	1.200	0.513–2.810	0.674			
CKD						
Absent	1					
Present	1.230	0.438–3.480	0.691			
COPD						
Absent	1					
Present	1.890	0.765–4.700	0.168			
Diabetes						
Absent	1					
Present	1.120	0.403–3.140	0.822			
SMI						
High-SMI	1					
Low-SMI	0.677	0.284–1.620	0.380			
Visceral fat area						
<100 cm^2^/m^2^	1					
≥100 cm^2^/m^2^	0.936	0.391–2.240	0.882			
GLIM-defined malnutrition						
Absent	1					
Present	0.723	0.295–1.770	0.478			

## Data Availability

The datasets generated and/or analyzed during the current study are available upon reasonable request from the corresponding author.

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
