# Peer review of "Preoperative Chronic Inflammation Is a Risk Factor for Postoperative Complications Independent of Body Composition in Gastric Cancer Patients Undergoing Radical Gastrectomy"

_cancers, 2024, doi:10.3390/cancers16040833_

Round 1

Reviewer 1 Report

Comments and Suggestions for Authors

Overall, the work by  Matsui et al.  concerns a very important topic, that is the characterization of preoperative inflammatory conditions that may predict post-gastrectomy complications. There are some points that need to be improved. Briefly, §Introduction and §Discussion should be improved. They are both too much short and they lack of bibliographic references. Data and statistics are interesting but their discussion must be improved. The authors focussed on “chronic”  inflammation and excluded the acute one. The term “inflammation” they used is too much general. Systemic inflammatory markers should be better introduced within the context of gastric cancer.. CRP, a well-known inflammatory marker, is a serum acute phase protein, induces TF expression and is a possible risk marker for venous thromboembolism. CRP enhances activation of inflammatory response(s) and coagulation system. Inflammation and coagulation are known to interact among each others, and, in particular, coagulation/fibrinolysis are known to be impaired/affected by GC. Since all work is focussed on CRP, these CRP different roles should be mentioned. CRP is known to predict coronary heart disease and arterial vascular events, is generally associated to tumor mass, and is induced by infiltrating lymphocytes/IL-producing monocytes. Among other inflammatory markers used to evaluate inflammation in GC patients there are also IL-6 and IL-23.

- Bibliography is not sufficiently cited and can be improved.

- The possible predictive role of preoperative inflammation after gastrectomy in GC-affected patients have been investigated by previous works. So, authors should better compare their findings with to those from other’s works. In particular, there is one published last year: doi: 10.21037/jgo-22-675. Among others, Imai et al (2022) found that “ Presepsin levels on PODs 3, 5, and 7 after gastrectomy is a more useful biomarker of postoperative infectious complications compared to CRP, WBCs, and Neuts, with a high sensitivity and specificity” [doi: 10.1038/s41598-022-24780-8]. The possible diagnostic accuracy of serum c-reactive protein (CRP) in predicting the development of postgastrectomy complications has been also investigated together with creatine phosphokinase (CPK) by Okubo et al (2021) [doi: 10.1186/s12885-021-07801-z]. Yu et al found that “increased pretreatment serum CRP level (≥10mg/L) was significantly associated with poor prognosis in gastric cancer patients, either in early or advanced stages” [doi: 10.7314/apjcp.2013.14.10.5735].

Line 100: authors should detail the post-operative complications with some examples and cite a Ref for the CD classification. The use of CD classification in evaluating the complications should be better explained.

Line 109-110 authors should explain how the PSM works and why they used it.

Line 186: preoperative inflammation estimated as CRP levels.

Line 190: where ? Add a Ref.

Line 203: such as anti-inflammatory agents or immunomodulatory nutritional supplements”, please add examples and Ref.

Line 204: preoperative chronic inflammation.

Line 206: the persistence of inflammation as in chronic inflammation (specify this otherwise it is not clear your thinking).

Are there any works analysing “neutrophil activation, and systemic inflammation through the secretion of proinflammatory cytokines” to cite ? Instead of a review, cite research works.

Line 229-231: please resentence. What do the authors mean with “immunosuppression”?

Comments on the Quality of English Language

There are some minor errors.

Author Response

Overall, the work by  Matsui et al.  concerns a very important topic, that is the characterization of preoperative inflammatory conditions that may predict post-gastrectomy complications. There are some points that need to be improved. Briefly, §Introduction and §Discussion should be improved. They are both too much short and they lack of bibliographic references. Data and statistics are interesting but their discussion must be improved. The authors focussed on “chronic”  inflammation and excluded the acute one. The term “inflammation” they used is too much general. Systemic inflammatory markers should be better introduced within the context of gastric cancer.. CRP, a well-known inflammatory marker, is a serum acute phase protein, induces TF expression and is a possible risk marker for venous thromboembolism. CRP enhances activation of inflammatory response(s) and coagulation system. Inflammation and coagulation are known to interact among each others, and, in particular, coagulation/fibrinolysis are known to be impaired/affected by GC. Since all work is focussed on CRP, these CRP different roles should be mentioned. CRP is known to predict coronary heart disease and arterial vascular events, is generally associated to tumor mass, and is induced by infiltrating lymphocytes/IL-producing monocytes. Among other inflammatory markers used to evaluate inflammation in GC patients there are also IL-6 and IL-23.

  1. Bibliography is not sufficiently cited and can be improved.

→Thank you for pointing this out. We have added more references in Introduction and Discussion.

  1. The possible predictive role of preoperative inflammation after gastrectomy in GC-affected patients have been investigated by previous works. So, authors should better compare their findings with to those from other’s works. In particular, there is one published last year: doi: 10.21037/jgo-22-675. Among others, Imai et al (2022) found that “Presepsin levels on PODs 3, 5, and 7 after gastrectomy is a more useful biomarker of postoperative infectious complications compared to CRP, WBCs, and Neuts, with a high sensitivity and specificity” [doi: 10.1038/s41598-022-24780-8]. The possible diagnostic accuracy of serum c-reactive protein (CRP) in predicting the development of postgastrectomy complications has been also investigated together with creatine phosphokinase (CPK) by Okubo et al (2021) [doi: 10.1186/s12885-021-07801-z]. Yu et al found that “increased pretreatment serum CRP level (≥10mg/L) was significantly associated with poor prognosis in gastric cancer patients, either in early or advanced stages” [doi: 10.7314/apjcp.2013.14.10.5735].

→Thank you for pointing this out. The paper you showed describes the relationship between postoperative inflammation and postoperative outcomes. This study examined the association between preoperative inflammation and postoperative outcomes. We cited the papers you presented in Discussion.

  1. Line 100: authors should detail the post-operative complications with some examples and cite a Ref for the CD classification. The use of CD classification in evaluating the complications should be better explained.

→Thank you for pointing this out. We have added references.

  1. Line 109-110 authors should explain how the PSM works and why they used it.

→Thank you for pointing this out. We added the following sentence:

“We categorized patients into inflammation and non-inflammation groups and adopted propensity score matching (PSM) with the aim of eliminating confounding factors in the comparison of various postoperative outcomes.”

  1. Line 186: preoperative inflammation estimated as CRP levels.

→Thank you for pointing this out. We revised the following:

“preoperative mild inflammation with CRP levels ≥ 0.5 mg/dL”

  1. Line 190: where ? Add a Ref.

→Thank you for pointing this out. We added the following sentence:

“Many facilities in Japan consider more than 0.5 mg/dL to be an abnormal laboratory value.”

  1. Line 203: such as anti-inflammatory agents or immunomodulatory nutritional supplements”, please add examples and Ref.

→Thank you for pointing this out. We revised the following:

“For advanced cancer patients with inflammation, anti-inflammatory agents such as non-steroidal anti-inflammatory drugs (NSAIDs) or aspirin, as well as immunomodulatory nutritional supplements like n-3 fatty acids, may be necessary for inflammation control [4].”

  1. Line 204: preoperative chronic inflammation.

→Thank you for pointing this out. We revised it as you indicated.

  1. Line 206: the persistence of inflammation as in chronic inflammation (specify this otherwise it is not clear your thinking).

→Thank you for pointing this out. We revised it as you indicated.

  1. Are there any works analysing “neutrophil activation, and systemic inflammation through the secretion of proinflammatory cytokines” to cite? Instead of a review, cite research works.

→Thank you for pointing this out. We did not find any research studies.

  1. Line 229-231: please resentence. What do the authors mean with “immunosuppression”?

→Thank you for pointing this out. We revised the following:

“The former may result from immunosuppression due to persistent chronic inflammation”

Reviewer 2 Report

Comments and Suggestions for Authors

The paper reports the results of a study to investigate if preoperative chronic inflammation is a risk factor for postoperative complications in pts with GC submitted to total gastrectomy.

1. The study is single-institutional and retrospective, and covers a long period of time (10 Yrs), this could lead to significant bias.

2. Preoperative "chronic inflammation" is defined only by having CPR levels >0.5 mg/dl, without any indications on how it reflects in the clinical status of the patients.

3. Title and conclusions state that "preop chronic inflammation is a risk factor for postoperative complications in pts with GC...". Materials and methods should refer only to the 2 groups of patients after matching comparison (99 vs. 99), and not also in all patients, where only 10% of the patients were in the group with CRP levels >0.5 mg/dl, and this group the patients were significantly older, had a more advanced clinical stage, underwent open surgery more frequently and had a higher Total Gastrectomy rate.

4. Preoperative inflammation was not a significat risk factor for developing  none of the complications considered (infectious, pneumonia, incisional SSI, intrabdominal abscess, pancreatic fistula, anastomotic leakage and severe complications), but only for the total amount of postop complications.

5. The design and the results of the study should be revised and reconsidered. The title and conclusions, also considering the biases of the study, should be less "sharp".

Author Response

The paper reports the results of a study to investigate if preoperative chronic inflammation is a risk factor for postoperative complications in pts with GC submitted to total gastrectomy.

  1. The study is single-institutional and retrospective, and covers a long period of time (10 Yrs), this could lead to significant bias.

→Thank you for pointing this out. To eliminate bias in retrospective study we used propensity score matching to make comparisons. This allows comparisons to be made as in a prospective RCT. Multivariate analysis is also performed to further prove universality. We included this time period because postoperative management did not change within this study period.

  1. Preoperative "chronic inflammation" is defined only by having CPR levels >0.5 mg/dl, without any indications on how it reflects in the clinical status of the patients.

→Thank you for pointing this out. We believe it is clinically meaningful to be able to detect risk factors for postoperative complications using only a single indicator, CRP.

  1. Title and conclusions state that "preop chronic inflammation is a risk factor for postoperative complications in pts with GC...". Materials and methods should refer only to the 2 groups of patients after matching comparison (99 vs. 99), and not also in all patients, where only 10% of the patients were in the group with CRP levels >0.5 mg/dl, and this group the patients were significantly older, had a more advanced clinical stage, underwent open surgery more frequently and had a higher Total Gastrectomy rate.

→Thank you for pointing this out. They are confounding factors, and we performed a multivariate analysis in addition to examining them using propensity scores. Our results showed that preoperative chronic inflammation was a risk factor independent of these confounders.

  1. Preoperative inflammation was not a significat risk factor for developing  none of the complications considered (infectious, pneumonia, incisional SSI, intrabdominal abscess, pancreatic fistula, anastomotic leakage and severe complications), but only for the total amount of postop complications.

→Thank you for pointing this out. The lack of statistically significant differences is due to sample size, but the number of infectious complications and anastomotic leakages is twice as high as the number of occurrences, which we consider a clinically meaningful difference.

  1. The design and the results of the study should be revised and reconsidered. The title and conclusions, also considering the biases of the study, should be less "sharp".

→Thank you for pointing this out. Could you please explain what you intend by less "sharp"? We are stating the fact of the result, not a leap of faith.

Reviewer 3 Report

Comments and Suggestions for Authors

Thank you for the invitation to review this important manuscript. Excellent idea for the study. We are clinically aware of this issue but have never scientifically proven it. Here are my comments and suggestions.

When one looks at the table with complications (adjusted 99 / 99 patients), not a single complication has a statistically significant difference, but when analyzing total complications, there is a statistical difference. In such situations, can we say that there is a difference?

I agree with chronic inflammation from the primary tumor. But where are chronic inflammatory conditions removed, such as chronic bronchitis, periodontitis, caries, etc?

The degree of inflammation is also very important. I know having enough patients for subgroup analysis is difficult, but can you try doing at least two groups? Here is one of the scales:

Less than 0.3 mg/dL: Normal (level seen in most healthy adults).

0.3 to 1.0 mg/dL: Normal or minor elevation (can be seen in obesity, pregnancy, depression, diabetes, common cold, gingivitis, periodontitis, sedentary lifestyle, cigarette smoking, and genetic polymorphisms).

1.0 to 10.0 mg/dL: Moderate elevation (Systemic inflammation such as RA, SLE, or other autoimmune diseases, malignancies, myocardial infarction, pancreatitis, bronchitis).

More than 10.0 mg/dL: Marked elevation (Acute bacterial infections, viral infections, systemic vasculitis, major trauma).

More than 50.0 mg/dL: Severe elevation (Acute bacterial infections).

Author Response

Thank you for the invitation to review this important manuscript. Excellent idea for the study. We are clinically aware of this issue but have never scientifically proven it. Here are my comments and suggestions.

  1. When one looks at the table with complications (adjusted 99 / 99 patients), not a single complication has a statistically significant difference, but when analyzing total complications, there is a statistical difference. In such situations, can we say that there is a difference?

→Thank you for pointing this out. The lack of statistically significant differences is due to sample size, but the number of infectious complications and anastomotic leakages is twice as high as the number of occurrences, which we consider a clinically meaningful difference.

  1. I agree with chronic inflammation from the primary tumor. But where are chronic inflammatory conditions removed, such as chronic bronchitis, periodontitis, caries, etc?

→Thank you for pointing this out. As you point out, it is not possible to determine why chronic inflammation is occurring. We have added this to the study limitation.

“Thirdly, we cannot identify the cause of chronic inflammation. We cannot rule out other effects of chronic disease besides those arising from the cancer itself.”

  1. The degree of inflammation is also very important. I know having enough patients for subgroup analysis is difficult, but can you try doing at least two groups? Here is one of the scales:

Less than 0.3 mg/dL: Normal (level seen in most healthy adults).

0.3 to 1.0 mg/dL: Normal or minor elevation (can be seen in obesity, pregnancy, depression, diabetes, common cold, gingivitis, periodontitis, sedentary lifestyle, cigarette smoking, and genetic polymorphisms).

1.0 to 10.0 mg/dL: Moderate elevation (Systemic inflammation such as RA, SLE, or other autoimmune diseases, malignancies, myocardial infarction, pancreatitis, bronchitis).

More than 10.0 mg/dL: Marked elevation (Acute bacterial infections, viral infections, systemic vasculitis, major trauma).

 More than 50.0 mg/dL: Severe elevation (Acute bacterial infections).

→Thank you for your very important point. Because this study included patients undergoing elective gastrectomy, we excluded patients with acute inflammation. Therefore, it is difficult to group by CRP and the sample size is inadequate. We will increase the number of cases in the future.

Reviewer 4 Report

Comments and Suggestions for Authors

I read with interest the manuscript by Matsui et al. titled ‘Preoperative chronic inflammation is a risk factor for postoperative complications independent of body composition in gastric cancer patients undergoing radical gastrectomy’. The study investigates the possible association between preoperative inflammation and post-operative outcomes. I have the following remarks reported in a point-by-point manner:

1.     Given the relatively straightforward main outcome, the authors should provide a clearer definition of the collected measures, such as comorbidities and post-operative complications. In a retrospective trial spanning 10 years, the definition of complications is crucial as it may vary over time.

2.     In Table 4, labeled 'Results of analysis of risk factors for postoperative complications in patients without chronic inflammation,' the authors introduce the concept of chronic inflammation. It appears that the authors assume patients with CRP levels outside the specified range have chronic inflammation. It would be more accurate to state that CRP levels could indicate any state of inflammation, whether acute or chronic, and both may impact the post-operative course. I recommend revising this terminology throughout the manuscript.

3.     While the authors use CRP levels to reflect the state of inflammation, it's important to note that higher CRP levels do not unequivocally indicate an inflammatory status. Therefore, the authors should exercise caution in drawing conclusions throughout the discussion.

4.     The sentence 'As this study excluded patients with preoperative acute inflammation, the inflammation in this study may reflect the inflammation caused by cancer-induced cachexia' lacks sufficient evidence and should be removed.

5.     The sentence 'Increased inflammation immediately postoperatively has been reported to be associated with a higher incidence of postoperative complications' may be misleading. Postoperative complications are not necessarily a consequence of higher CRP levels but rather reflect the macroscopic events resulting from elevated CRP levels, which represent the microscopic side of the same event. I recommend reviewing and revising the discussion accordingly. Please refer to the article by Mari G, et al. Surg Laparosc Endosc Percutan Tech. 2016 Dec;26(6):444-448. doi: 10.1097/SLE.0000000000000324. PMID: 27783027 for further insights.

Author Response

I read with interest the manuscript by Matsui et al. titled ‘Preoperative chronic inflammation is a risk factor for postoperative complications independent of body composition in gastric cancer patients undergoing radical gastrectomy’. The study investigates the possible association between preoperative inflammation and post-operative outcomes. I have the following remarks reported in a point-by-point manner:

  1. Given the relatively straightforward main outcome, the authors should provide a clearer definition of the collected measures, such as comorbidities and post-operative complications. In a retrospective trial spanning 10 years, the definition of complications is crucial as it may vary over time.

→Thank you for pointing this out. We have added definitions for outcomes and comorbidities.

  1. In Table 4, labeled 'Results of analysis of risk factors for postoperative complications in patients without chronic inflammation,' the authors introduce the concept of chronic inflammation. It appears that the authors assume patients with CRP levels outside the specified range have chronic inflammation. It would be more accurate to state that CRP levels could indicate any state of inflammation, whether acute or chronic, and both may impact the post-operative course. I recommend revising this terminology throughout the manuscript.

→Thank you for pointing this out. This study reveals that preoperative chronic inflammation is a risk factor for postoperative complications, and it notes that no additional risk factors were detected in patients with preoperative chronic inflammation. On the other hand, in patients without preoperative chronic inflammation, traditional risk factors were identified. Thus, it indicates that the presence or absence of chronic inflammation determines different sets of risk factors.

  1. While the authors use CRP levels to reflect the state of inflammation, it's important to note that higher CRP levels do not unequivocally indicate an inflammatory status. Therefore, the authors should exercise caution in drawing conclusions throughout the discussion.

→Thank you for pointing this out. As you rightly pointed out, we have mentioned the inability to determine the duration of sustained inflammation as a study limitation.

“Fourthly, the preoperative inflammatory status is assessed at a single time point, and there is no mention of the duration of sustained inflammation.”

  1. The sentence 'As this study excluded patients with preoperative acute inflammation, the inflammation in this study may reflect the inflammation caused by cancer-induced cachexia' lacks sufficient evidence and should be removed.

→Thank you for pointing this out. I added references as suggested by another reviewer, who recommended adding citations to the same sentence.

“As this study excluded patients with preoperative acute inflammation, the inflammation in this study may reflect the inflammation caused by cancer-induced cachexia [19, 20].”

  1. The sentence 'Increased inflammation immediately postoperatively has been reported to be associated with a higher incidence of postoperative complications' may be misleading. Postoperative complications are not necessarily a consequence of higher CRP levels but rather reflect the macroscopic events resulting from elevated CRP levels, which represent the microscopic side of the same event. I recommend reviewing and revising the discussion accordingly. Please refer to the article by Mari G, et al. Surg Laparosc Endosc Percutan Tech. 2016 Dec;26(6):444-448. doi: 10.1097/SLE.0000000000000324. PMID: 27783027 for further insights.

→Thank you for pointing this out. What we intended to convey is excessive inflammation specifically within the context of postoperative high inflammation. We have made the following correction.

“In previous reports, excessive postoperative inflammation has been documented to increase the incidence of postoperative complications [5, 10-13]”

Round 2

Reviewer 1 Report

Comments and Suggestions for Authors

Accordingly to the revisions, this version of the manuscript is improved.
All the points I have addressed to the Authors have been correctly revised.

Reviewer 3 Report

Comments and Suggestions for Authors

All queries answered. No additional comments.